# Infill Design Reinforcement of 3D Printed Parts Using Refinement Technique Adapted to Continuous Extrusion

**Sashi Kiran Madugula \***(ID)**, Laurence Giraud-Moreau, Pierre-Antoine Adragna** (ID) **and Laurent Daniel** (ID)

ICD-LASMIS, Université de Technologie de Troyes, 12 Rue Marie Curie, CS 42060, 10004 Troyes CEDEX, France; laurence.moreau@utt.fr (L.G.-M.); pierre_antoine.adragna@utt.fr (P.-A.A.); laurent.daniel@utt.fr (L.D.)
\*  Correspondence: sashi-kiran.madugula@utt.fr or sashikiran08@gmail.com

**Abstract:** In this paper, we introduce an advanced numerical tool aimed to optimise the infill design of 3D printed parts by reducing printing time. In 3D printing, the term infill refers to the internal structure of a part. To create the infill design, slicing software is used, which generally creates the infill uniformly throughout the part. When such a part is subjected to external loading, all the infill regions will not experience the same amount of stress. Therefore, using uniform infill throughout the part is not the most optimised solution in terms of material usage. We do propose to evolve the infill design with respect to the mechanical stresses generated by the external loads. To achieve this, an advanced numerical tool has been developed, based on refinement techniques, to control the infill design. This tool is coupled with Finite Element Simulation (FE Simulation) software, which helps to identify the zones where the material is required as an infill to reinforce a part, whereas the refinement technique makes it possible to place the material as an infill in such a way that the airtime during the printing of the part is zero. Zero airtime printing is defined as the ability to deposit each layer of a part, without stopping the material extrusion during the displacement of the nozzle. Therefore, the proposed numerical tool guides us to generate the infill design of a part, in such a way that it will consume zero airtime while manufacturing. Simultaneously, it will increase the stiffness of the part. The proposed approach is here applied to a rectangular structure subjected to four-point bending, made up of PLA material (Poly-Lactic Acid).

**Keywords:** infill structure; continuous printing; additive manufacturing; refinement; finite element simulation (FE Simulation); fused deposition modelling (FDM)

## 1. Introduction

Additive manufacturing began to emerge in the 1980s with stereolithography and makes it possible to manufacture a part layer by layer. Today, this method has become a very interesting alternative to conventional manufacturing processes and is particularly well suited in the case of single parts, small series or complex geometry shapes such as lattice structures [1]. Amongst the various additive manufacturing processes, the most commonly used, Ref. [2] and the most affordable is the hot wire deposition process, called FDM (Fused Deposition Modeling) or FFF (Fused Filament Fabrication). It consists of depositing a thermoplastic material on a support in order to build a part layer by layer. The material is packaged in the form of a coil of wire or granulates. Material used for this process are thermos fusible polymers like Poly-Lactic Acid (PLA), Acrylonitrile Butadiene Styrene (ABS), Polyethylene Terephthalate Glycol (PETG), and Acrylonitrile Styrene Acrylate (ASA).

In recent years, most research about the FDM process has focused on the reduction of the manufacturing time or the increase of mechanical stiffness. Despite its name of rapid prototyping, additive manufacturing by hot wire deposition (FDM) is a relatively slow process. Numerous studies have been conducted in order to reduce FDM manufacturing time. Some researchers have proposed deposition strategies to reduce nozzle displacement [3] or to maximise the feed speeds during printing [4]. Yu-An, Yon et al. [5] suggest

locally producing the part on several successive layers before changing the printing zone. Another way to reduce manufacturing time is continuous printing. The aim of continuous printing is to drive the airtime to zero. Without it, around 25% of the nozzle travel distance are displacements without extrusion. Moreover, these kinds of material depositions can cause local heterogeneity in the manufactured part, and therefore can lead to fragility against mechanical loading. However, to attain continuous printing, a constraint should be satisfied, and a path that visits each node only once shall be generated [6,7]: This is also called the Chinese Postman Problem. Very few studies have been proposed so far connecting this problem with FDM manufacturing process.

The FDM process offers the capability to produce parts of complex external shapes, but also complex internal geometries (infill) such as cellular or lattice structures. The main benefit of using a lattice structure is to decrease both material and time consumption while keeping the strength of the part constant, thus resulting in cost reductions. The infill design is generally created by slicing software and it is generally uniform throughout the structure. Infill percentage influences the printing time, the weight of the part and also the mechanical stiffness of the part. Optimising the strength-to-weight ratio is therefore one of the main objectives of the research. Several studies have investigated the effect of different regular infill patterns and infill densities on mechanical strength [8,9]. Ei Cho et al. [10] investigated several infill patterns: zigzag, grid and triangle. Among these last three, triangle pattern has given the highest strength. As these infill designs are kept uniform throughout the part, localised mechanical stresses are not addressed. Furthermore, Ahn et al. [11] have shown that specimens fabricated by fused deposition modelling display anisotropic behaviour and are significantly influenced by the orientation of the layered raster.

Using the results of finite element simulation to guide the filament trajectory is another approach to improve the mechanical properties of the FDM printing part. Gopsill and Hicks [12], J. Gardan et al. [13] and Madugula et al. [14] have shown that the alignment of the filament trajectory in the direction of the principal stress can improve the mechanical properties of the part. Similar work has been done by Kwock et al. [15] and Wu et al. [16]. Gopsill and Hicks [12] have proved that around 76% more stiffness of the part can be obtained with the alignment of the filament in the direction of the principal stress. This proved that placing material in the direction of principal stress truly could help to reduce material usage and increase the stiffness of the part. In another study, the same authors [17] have proposed a method using the results of finite element analysis (FEA) to influence the internal geometry of components. A rectangular part undergoing different loading scenarios has been studied. The part with optimised infill was then compared to a part containing uniform honeycomb infill. Results have shown an increase in the stiffness of the optimised part. Here, a FE Simulation was used just as a guide to define the final infill design.

In addition, S. I. Park et al. [18] and W. Chen et al. [19] developed lattice cells for the internal structures of the part, which is a very strategical approach for reducing the material usage, whereas a more innovative technique was adopted by L. Cheng et al. [20] and W. Chen et al. [19]. They have worked on a variable cross-sectional area of lattice infill depending upon the stresses undergone. However, this approach does not take into account the constraints induced while really printing variable sized lattice structure, using the FDM process, and also does not present the ability to print the part without having an airtime.

In the literature, many studies were conducted on topological optimisation. The purpose of topological optimisation is to find the most beneficial material deposition by changing the contour of the part. Whereas, infill optimisation addresses the same idea of the best material deposition while keeping the contour of the part constant. For this, many studies have been carried out using lattice structures [19,20]. However, the importance of printing parameters was not taken into consideration while optimising infill. Therefore, our goal was to propose the infill optimisation technique taking into account the printing

parameters such as minimum printable infill size possible and to reduce the manufacturing time of the final part using continuous extrusion.

In this paper, we would like to introduce an iterative process, coupled with FE Simulation software and a refinement technique that will both increase the stiffness of the part and drive to zero airtime. In our approach, the contour of the part is assumed to always be constant. This paper presents the different steps of the proposed method. This proposed approach is implemented on a rectangular part of a constant section, subjected to four-point bending forces, in order to evaluate and validate the proposed method. The part with optimised infill design obtained by the refinement procedure is then compared to the part with uniform infill designs. Results are discussed with some possible future works before coming to the conclusion.

## 2. Methodology

### 2.1. Continuous Printing

Continuous printing consists of depositing each layer of the part without stopping the extrusion during nozzle displacements. Our primary aim of using continuous extrusion printing is time-saving. Printing the part in a single continuous path also avoids the superimposition of filaments, which often distorts the final printed part. In this paper, some results of graph theory are used to generate a path that visits each node only once. A print path respecting such a constraint of passage must be the solution to the Chinese Postman Problem. This then implies that the path must form an Eulerian or semi-Eulerian graph to enable continuous printing.

For a graph to be Eulerian, each node must have an even number of branches. In such a case, if any node of the graph is a starting point of the path, then necessarily it will also be the end of the path, as shown in Figure 1. Considering Figure 1 (left), if printing is started from node A, then printing can be ended at the same point A by passing through all the branches (AB, BC, CD & DA) only once. A graph is semi-Eulerian, as shown in Figure 1 (centre), if two of its nodes have an odd number of branches coming out of it (B and D). In this case, one of these two nodes (B or D) must be a starting point of the path and the other one will be the endpoint of the print path. Finally, a path is a non-Eulerian path, as shown in Figure 1 (right)), if more than two nodes have an odd number of branches. In this case, continuous printing is not possible. It is therefore important that the refinement method used to define the internal structure of the part allows forming an Eulerian or semi-Eulerian graph. The next subsection presents the iterative technique used to optimise the internal structure (infill design) as well as the integrated rules to allow continuous printing.

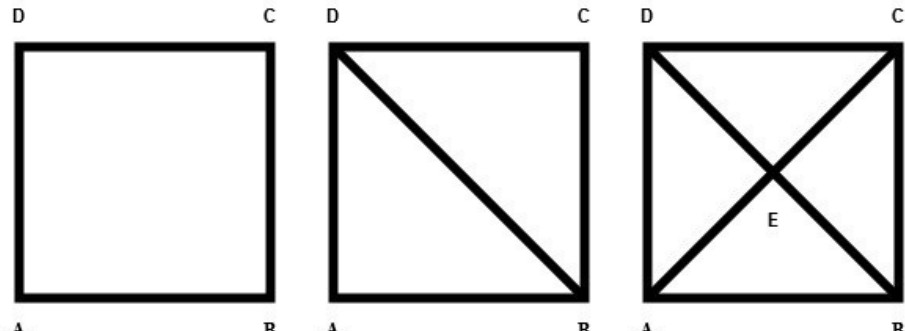

**Figure 1.** Eulerian path (**left**), semi-Eulerian path (**centre**), non-Eulerian path (**right**).

### 2.2. Describing Iterative Process

The proposed method is based upon an automatic iterative process, which integrates finite element simulation and refinement techniques. This process evolves automatically, without any user intervention. At each iteration, we take into account the new infill design of the part and refine it. Refinement rules are defined in order to guarantee that the optimised internal design leads to an Eulerian or semi-Eulerian path. It should be

noted that the refinement applies solely to the printed part infill structure. This refinement is purely geometrical and should not be confused with finite element mesh refinement. Here, the refinement means to make denser some areas of the internal geometry, to place more material, and thus to have in these zones an infill structure made of smaller cells. As a result, the printed part will be strengthened in previously detected high-stress areas. To practically achieve this, we designed a Matlab program allowing us to recreate the input file for Abaqus at each iteration.

The test part presented later in this paper has a constant cross-section, and 2D beam elements are used to mesh the part. That part is composed of a fixed contour and an infill structure that will evolve at each iteration. The mean steps of the proposed iterative method are presented in Figure 2. The proposed iterative method comprises the following seven main steps:

1. Coarse initial infill generation;
2. Input file generation for simulation;
3. Finite Element simulation;
4. Analyse results;
5. Test condition;
6. Infill refinement;
7. G-code generation for printing.

Further details of each step of the method are presented in the following subsections.

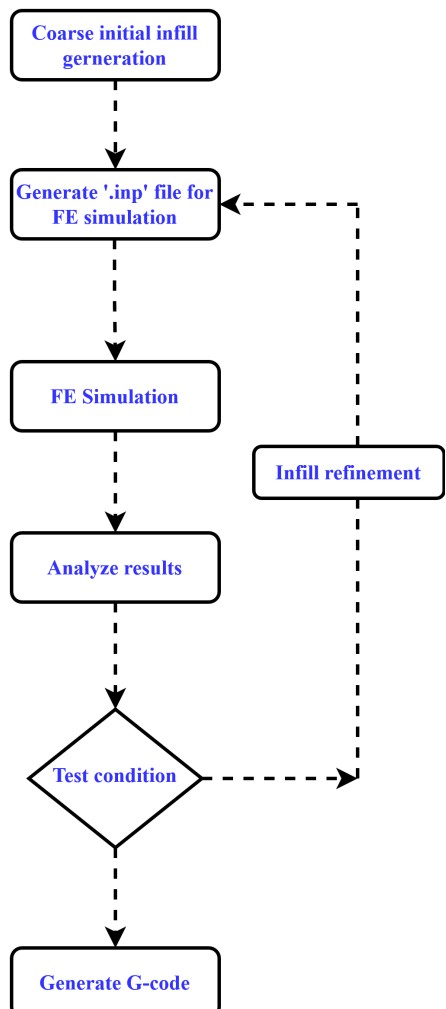

**Figure 2.** Iterative loop for the refinement technique.

### 2.2.1. Coarse Initial Infill Generation

The first step is the generation of the part with a coarse initial infill defined by the user. It is made up of uniformly distributed triangular or quadrangular patterns.

### 2.2.2. Generating Input File ('.inp')

This input file is based on fixed user inputs (e.g., dimensions of the part, material properties, applied forces, boundary conditions ... etc.) and includes the finite element mesh of the part for the FE Simulation.

### 2.2.3. Finite Element Simulation

The purpose of this calculation step is to determine part mechanical stresses and strains when subjected to an external load, for a given internal geometry. A Finite Element calculation is carried out with Abaqus. Two-dimensional beam elements (B21, Timoshenko beam) are used to mesh the part. An elastic model with implicit analysis is used such that the stresses do not exceed the elastic limit of the material. At the end of each finite element simulation, output result files containing stresses, strains, and displacements for all the mesh elements and nodes are generated by Abaqus.

### 2.2.4. Analyze Results

This step is to analyse finite element simulation results. Files generated by the previous step containing results values are used to identify elements that are not satisfying the stopping criteria. Stopping criteria are:

1. Threshold stress of the structure ($\sigma_{thrld}$).
2. Minimum length of a side of the infill design (L).

Structure threshold stress ($\sigma_{thrld}$) is a user-defined value, and infill design minimum size is based on a 3D printer parameter also defined by the user.

1. Threshold stress of the structure ($\sigma_{thrld}$).
   Structure threshold stress ($\sigma_{thrld}$) is the maximum von Mises stress value that the part is required to withstand. During the bending, each beam element is subjected to tension and compression stress (see Figure 3). Hence, the maximum absolute value between both the corresponding von Mises stresses ($\sigma_{vm}$) is taken into consideration. The FE Simulation (elastic model) results allow identifying the potential beams with a maximum absolute value of von Mises stress ($\sigma_{vm}$) above threshold limit ($\sigma_{thrld}$). Each infill design connected to those beams is then identified and will undergo the refinement process.
2. Minimum length of a side of the infill design (L)
   Infill side minimum length (L) is also considered as a stopping criterion due to the constraint of the 3D printer. This criterion depends on the nozzle diameter used for generating the part. In our case, to identify this critical length, several equilateral triangular structures have been printed of a side length varying from 1 mm to 2 mm using a nozzle diameter of 0.4 mm. Three-dimensional printed triangular structures with a side length less than 1.2 mm were found to be fully dense structures, whereas a side length of triangular infill design above or equal to 1.2 mm generates a structure with proper visibility, as shown in Figure 4. Therefore, for the refinement process, the minimum length (L) is set to twice the value of 1.2 mm previously found. Thus, after infill refinement, the possible length of each beam will always be greater than or equal to 1.2 mm.

Therefore, the refinement process will only be performed, if:

- Any beam corresponding to any infill design has a maximum absolute von Mises stress ($\sigma_{vm}$) > threshold stress ($\sigma_{thrld}$);
- Simultaneously, the length of the side of that corresponding infill (L) $\geq$ 2.4 mm.

Thus, iteration continues until any one of the stopping criteria is satisfied.

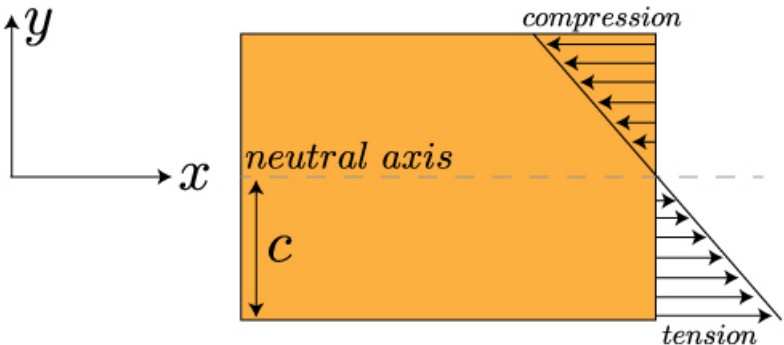

**Figure 3.** Stress distribution on the beam during bending.

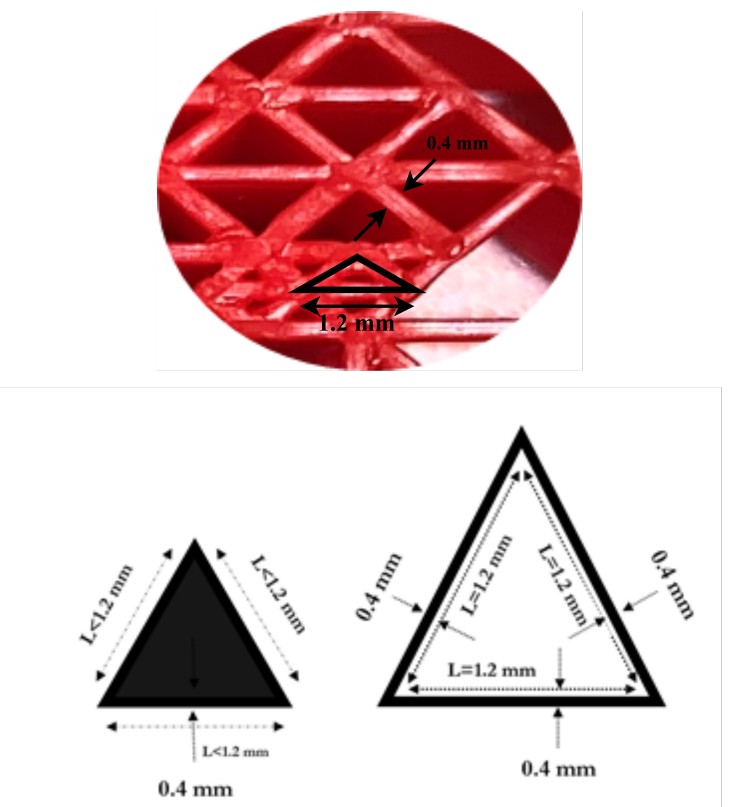

**Figure 4.** Size limitation of triangular infill design.

### 2.2.5. Test Condition

Before refinement, if stopping criteria has been attained throughout the whole infill of the structure, we exit from the iterative loop. If not, a refinement process will be conducted and a new '.inp' file is generated for the next finite element simulation. While exiting from the iterative loop, hopefully, we managed to get under maximum threshold stress everywhere. However, it is possible to get to a condition where the maximum von Mises stress ($\sigma_{vm}$) is still above threshold stress ($\sigma_{thrld}$) in some areas, while we cannot refine them anymore, because we reached the minimum length of a side of the infill (L) criterion there.

### 2.2.6. Refinement

This step involves the generation of a new infill design by using refinement procedures derived from re-meshing methods, generally used to improve the accuracy of FE calculations. The proposed refinement method consists of refining the infill design elements selected during the previous step, according to the stopping criteria. Here, refinement means adding walls inside the infill structure, and thus increasing the infill density.

To comply with continuous printing, defined refinement rules must guarantee an Eulerian or semi-Eulerian printing path. The very simple following rules are used to refine the infill:

- A triangle is subdivided into four triangles: an inscribed triangle and three triangles in the corners (see Figure 5).
- A quadrangle is subdivided into one quadrangle and four triangles: an inscribed quadrangle and four triangles in the corners (see Figure 5).

These refinement rules enable the infill geometry to satisfy continuous printing constraint. Indeed, the subdivision techniques proposed above allows us to form an Eulerian path with an even number of branches coming out of each node, so the new generated infill design can be printed in a single continuous path.

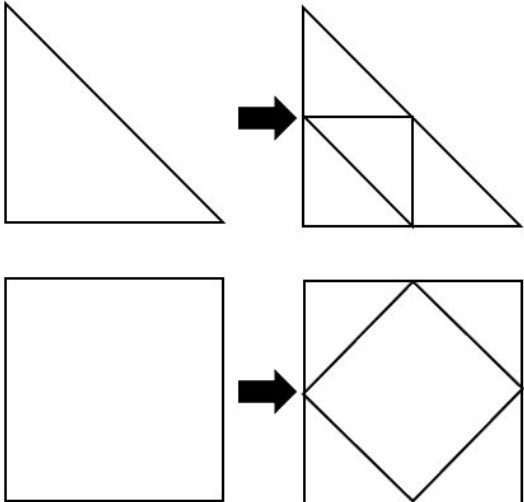

**Figure 5.** Refinement of triangular structure and quadrangular structure.

To guarantee better infill structure homogeneity, an additional refinement criterion has been added. It consists of a refining region surrounded by two adjacent refined regions. Figure 6 illustrates this principle. The two yellow regions have been refined. The white region with two refined adjacent regions is then also refined to obtain a more homogeneous infill.

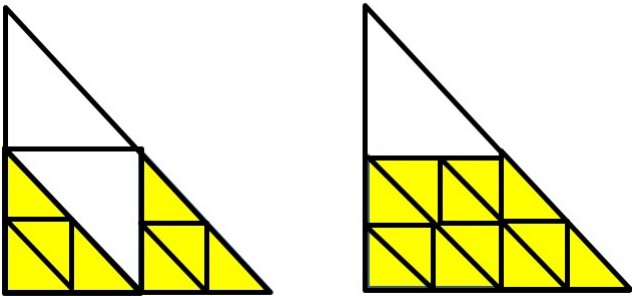

**Figure 6.** Refinement of infill design.

### 2.2.7. Generating G-Code

During the final step, Matlab function is used to generate the G-code file using the coordinates of the nodes, which will be readable by the 3D-printer, to produce the part. The whole part is then printed layer by layer. Thanks to the proposed refinement technique, the part can be printed in a single continuous path, without any airtime.

## 3. Numerical Simulation

To demonstrate the effectiveness of the proposed iterative method, a rectangular structure subjected to four-point bending has been established as a case study. It is presented in Figure 7. This rectangular structure is 180 mm in length, 30 mm in height and 20 mm in depth. Concentrated loads are applied on the top of the structure, separated by a distance of 90 mm. As boundary conditions, roller and fixed joint are implemented in the bottom right and bottom left side of the structure, respectively. The initial infill design of the case-study part is made of 14 triangular and 6 quadrangular structures. Each of the triangle and quadrangle side is assumed as a beam with a constant rectangular cross-sectional area of size '0.4 mm × 20 mm' as presented in Figure 7. The value of '0.4 mm' is the diameter size of the extrusion nozzle of RAISE3D printer, and '20 mm' represents the depth of the whole part.

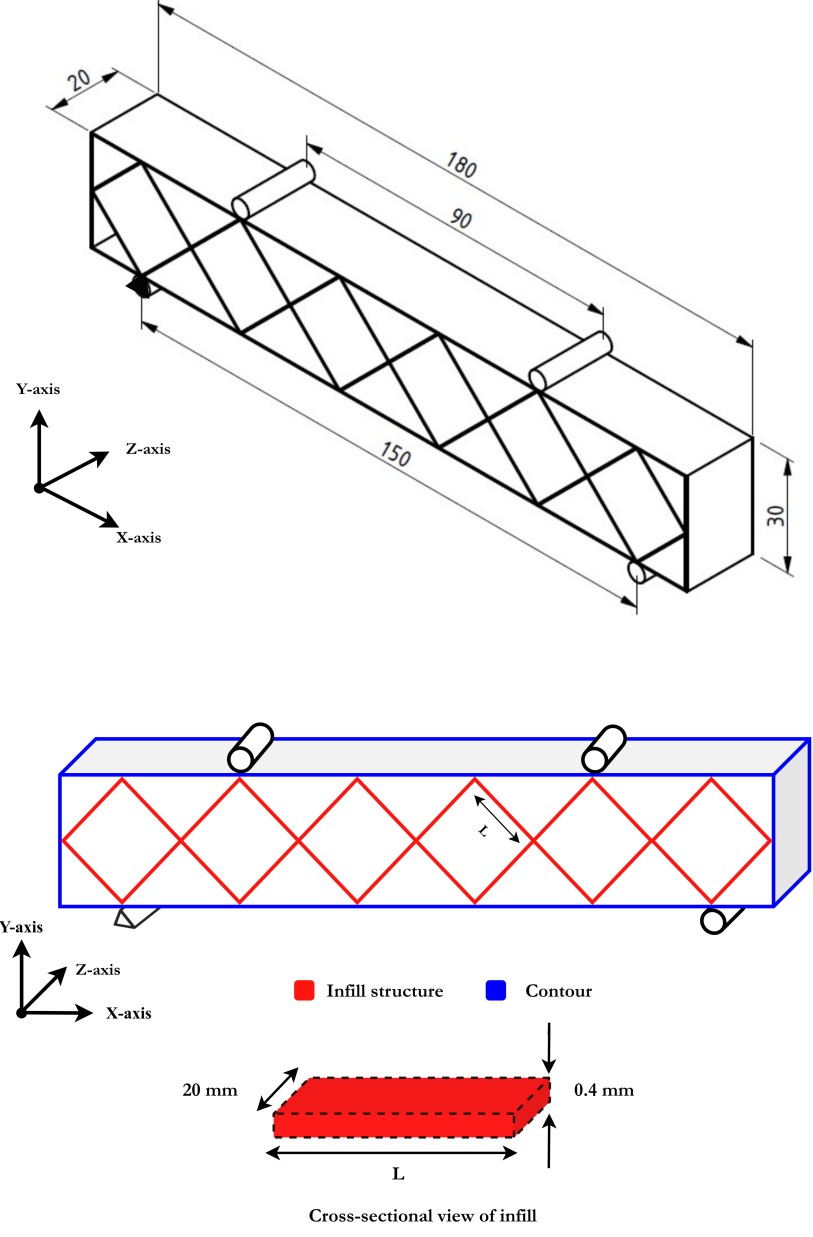

**Figure 7.** Case study.

### 3.1. Parameters for Numerical Simulations

Two-dimensional finite element simulations using an linear elastic constitutive model with implicit analysis are adopted in Abaqus. Thermoplastic PLA (Poly-Lactic Acid) is used as a material with flexural modulus of elasticity (E = 3.6 GPa), Poisson's ratio ($\mu$ = 0.35) and yield stress of 50 MPa. For finite elements calculations, each beam is subdivided into mesh elements (B21: Timoshenko beam) of size 0.1 mm. A concentrated load of 250 N is applied on each end to the structure subjected to four-point bending.

### 3.2. Simulation Results

The following sections present and discuss the results from the case study in relation. An initial test of the refinement technique using a threshold stress of 10 MPa, and a minimum infill design side length of 2.4 mm, came to a stop after six iterations as shown in Table 1. Von Mises stress distribution and displacements from Abaqus during each iteration are presented in Tables 2 and 3.

**Table 1.** Evolution of part infill design when subjected to a 4-point bending using threshold stress = 10 MPa and minimum length of a side of the infill design (L) $\geq$ 2.4 mm as stopping criteria.

| Iteration | Infill Design of Part |
|:---:|:---:|
| 1 |  |
| 2 |  |
| 3 |  |
| 4 |  |
| 5 |  |
| 6 |  |

**Table 2.** Figures of von Mises stress distribution for each iteration in Table 1 obtained in Abaqus.

| Iteration | Infill Design of Part |
|:---:|:---:|

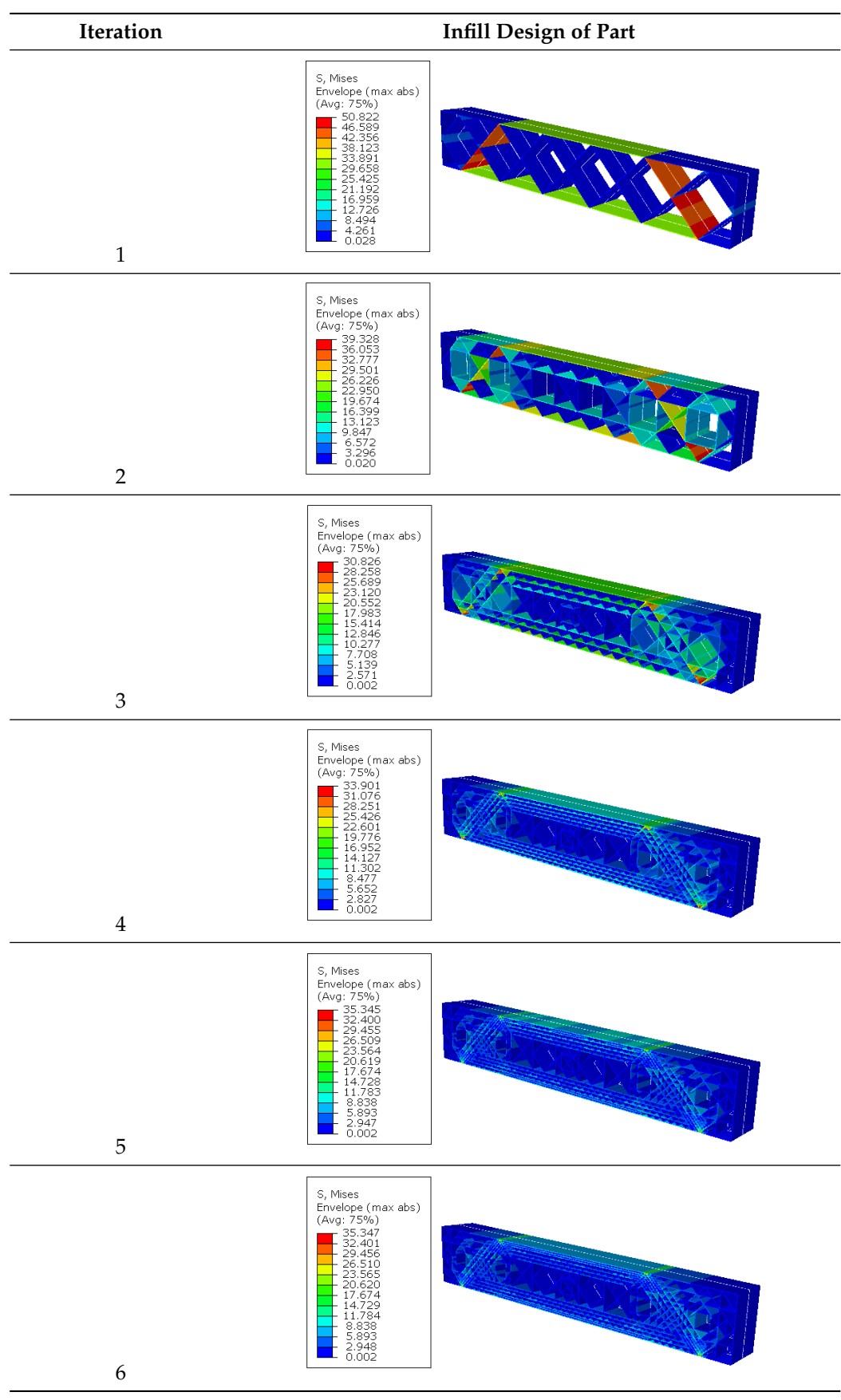

**Table 3.** Figures of maximum displacement for each iteration in Table 1 obtained in Abaqus.

| Iteration | Infill Design of Part |
|:---:|:---:|
| 1 | |
| 2 | |
| 3 | |
| 4 | |
| 5 | |
| 6 | |

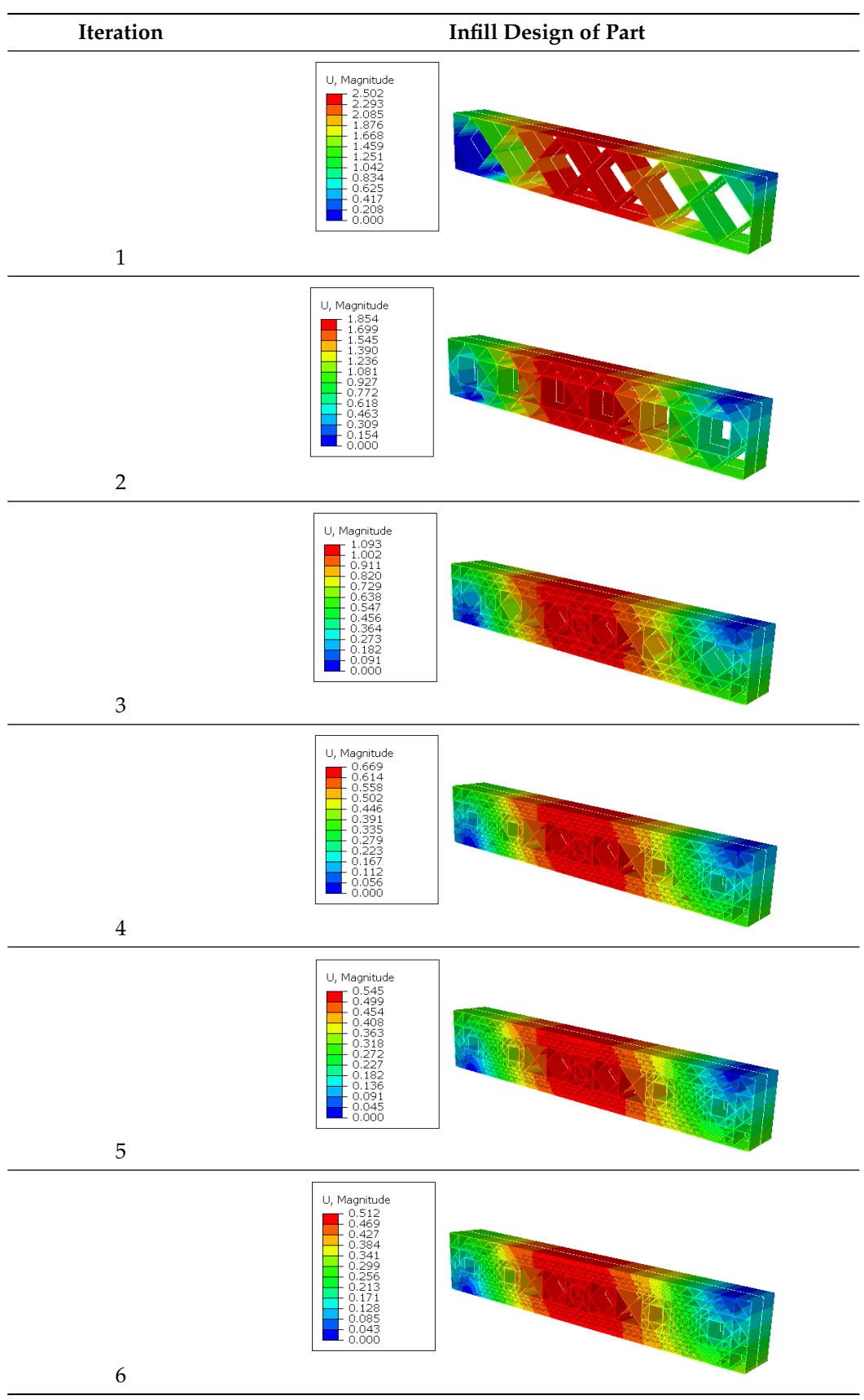

Observing the numerical simulation results in Table 4, it is obvious that the maximum absolute value of von Mises stress ($\sigma_{vm}$) = 36.42 MPa of the part after the final iteration, is still higher than threshold stress ($\sigma_{thrld}$) = 10 MPa. Further refinement could not be possible, because the minimum length ('L') constraint of 2.4 mm has been reached. This implies that all the sides of infill designs corresponding to those beams with ($\sigma_{vm}$) greater than ($\sigma_{thrld}$) have attained the criteria of minimum length ('L' $\leq$ 2.4 mm). Hence, iteration has finished. Table 4 also illustrates maximum displacement (mm) of the part, strain energy (mJ) and finally infill density in percentages after each iteration. The maximum displacement (mm) has plunged from the first iteration (2.5 mm) to the fourth iteration (0.66 mm) and remains almost constant throughout the two remaining iterations. Meanwhile, from the same table, it is clear that there has been a significant increase in the infill density from 6.8% to 41%, while the strain energy for the final part has decreased to 92.24 mJ by 78.7% from 433.35 mJ in the first iteration. Thus, making the part stiffer after every iteration.

**Table 4.** FE Simulation results for case-study subjected to 4-point bending with threshold stress = 10 MPa and minimum length of a side of the infill design (L) $\geq$ 2.4 mm as stopping criteria.

| Iteration | Max. Absolute of von Mises Stress ($\sigma_{vm}$ in MPa) | Max. Displacement ($\delta$ in mm) | Strain Energy (mJ) | Infill Density |
|---|---|---|---|---|
| 1 | 50 | 2.5 | 433.35 | 6.8% |
| 2 | 39.328 | 1.85 | 326 | 12% |
| 3 | 30.826 | 1.09 | 196 | 22% |
| 4 | 34 | 0.66 | 120.6 | 35% |
| 5 | 35.3 | 0.54 | 97.12 | 40% |
| 6 | 35.3 | 0.51 | 92.24 | 41% |

The final part with the infill design shown in Figure 8 after six iterations, was thus employed to create the G-code using Matlab code for the 3D-printer. This infill design of the final part satisfies the constraints of continuous printing with each nodes containing an even number of branches coming out of it.

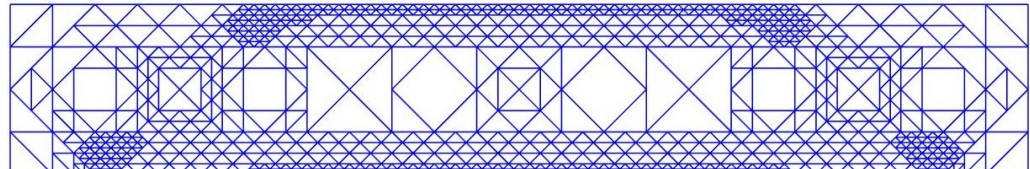

**Figure 8.** Final part with optimised infill using refinement technique with threshold stress = 10 MPa and minimum length of a side of the infill design (L) $\geq$ 2.4 mm as stopping criteria (infill density = 41%).

Same study was carried out again using threshold stress of 17.5 MPa and 25 MPa as stopping criteria. The stopping criteria of minimum length of the infill ('L' $\geq$ 2.4 mm) remains the same for both these cases due to the usage of the same 3D printer. It took six iterations to reach the final structure for both threshold stresses ($\sigma_{thrld}$). For threshold stress of 17.5 and 25 MPa, part with final infill design is presented in Figures 9 and 10, respectively. Whereas, evaluation of infill density percentage during each iteration for each threshold stress ($\sigma_{thrld}$) = 10 MPa, 17.5 MPa, and 25 MPa, is presented in Figure 11.

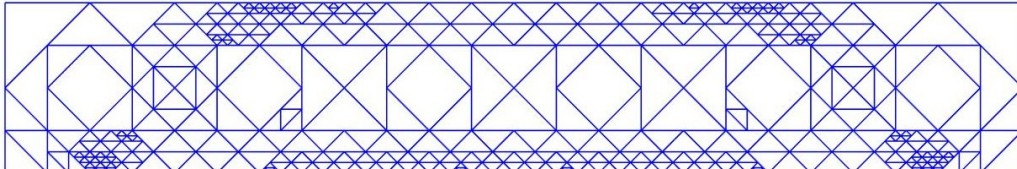

**Figure 9.** Final part with optimised infill using refinement technique with threshold stress = 17.5 MPa and minimum length of a side of the infill design (L) ≥ 2.4 mm as stopping criteria (infill density = 26%).

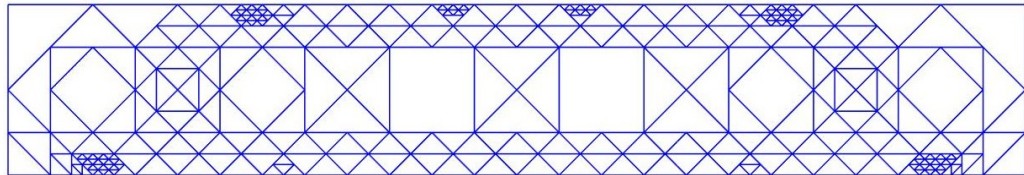

**Figure 10.** Final part with optimised infill using refinement technique with threshold stress = 25 MPa and minimum length of a side of the infill design (L) ≥ 2.4 mm as stopping criteria (infill density = 22%).

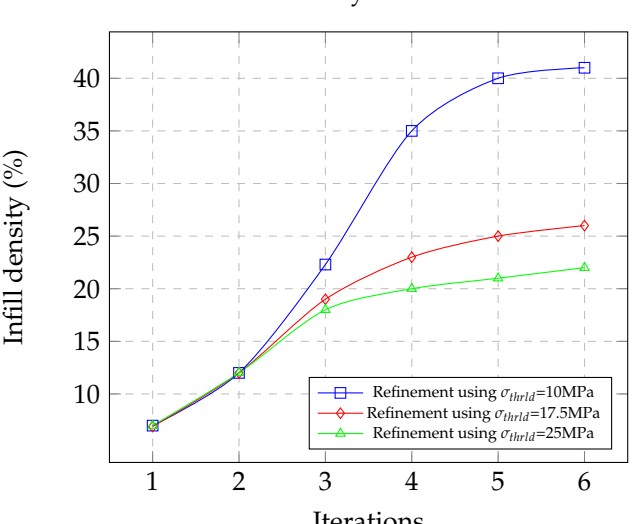

**Figure 11.** Infill density during each iteration.

## 4. Comparison

Infill density percentages of the final part generated after using refinement technique, with threshold stress as 10 MPa, 17.5 MPa and 25 MPa are 41%, 26% and 22%, respectively. For comparison, structures with uniform infill design with the same dimensions (180 × 30 × 20 mm) and the same infill density (41%, 26% and 22%) have been generated using BL2D software [21] as shown in Figures 12–14. It should be noted that these parts with uniform infill do not allow continuous printing. Table 5 presents the FE Simulation results of parts with uniform infill density (41%, 26% and 22%) using the same boundary and loading conditions as the one used for the optimised case study in the previous section. Corresponding G-codes for printing are generated using Matlab script Section 2.2.7. Tables 6–8 show comparison between parts with optimised and uniform infill design in terms of FE Simulations and printing time.

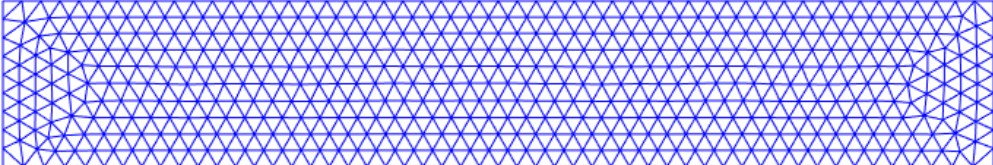

**Figure 12.** Part with uniform infill density of 41%.

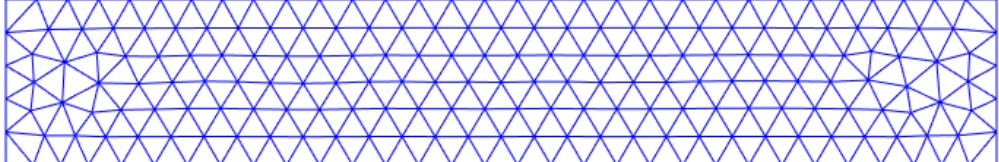

**Figure 13.** Part with uniform infill density of 26%.

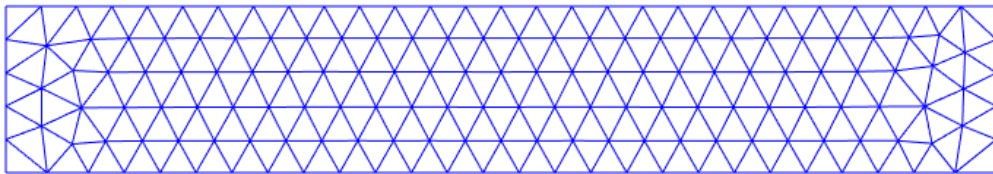

**Figure 14.** Part with uniform infill density of 22%.

**Table 5.** FE simulation results of part with uniform infill design.

| Uniform Infill Density (%) | von Mises Stress ($\sigma_{vm}$ in MPa) | Max. Displacement ($\delta$ in mm) | Strain Energy (mJ) |
|---|---|---|---|
| 41% | 23.5 | 0.8 | 141.31 |
| 26% | 25.1 | 1.21 | 205.1 |
| 22% | 26.7 | 1.33 | 225.9 |

### 4.1. Comparison of Parts with Same Infill Density of 41%

Focusing on Table 6, it is immediately apparent that for the same infill density of 41% and almost same filament weight usage, the infill of the part generated by the refinement process performs better than the part with uniform infill design in terms of displacement and strain energy. Comparing maximum displacement, structure with infill design generated by the refinement process has lower displacement value (0.51 mm) than the structure with uniform infill design (0.83 mm) by 35%. For a similar instance, if we compare strain energy for the optimised infill part, it is 92.24 mJ. While on the other side, the value is 141.31 mJ for the part with 41% uniform infill density. This highlights the importance of placing the material in the high-stress zones as infill compared to uniformly placing material. Finally, printing time for the part with optimised infill design consumes 5.11 h compared to it's counterpart with uniform infill in 7.01 h (see Figure 15). This is possible due to defined refinement rule for continuous printing.

Although, the maximum von Mises stress ($\sigma_{vm}$) value of the optimised infill design (35.3 MPa) is higher than the part with a uniform infill design (23.5 MPa). This is due to the fact that beam orientation in the part with uniform infill is better than the one in the part with optimised infill design.

**Table 6.** Comparison of the part with uniform infill to the part with infill defined by refinement process using threshold stress = 10 MPa and minimum length of a side of the infill design (L) $\geq$ 2.4 mm as stopping criteria.

|  | Figure 16 (Top) | Figure 16 (Bottom) |
| --- | --- | --- |
| **Infill density (%)** | **41** | **41** |
| Von Mises Stress (MPa) | 35.3 | 23.5 |
| Maximum Displacement (mm) | 0.51 | 0.8 |
| Elastic Strain Energy (mJ) | 92.2 | 141.31 |
| Total mass of the part (g) | 41.8 | 41 |
| Time to print (hours) | 5.11 | 7 |

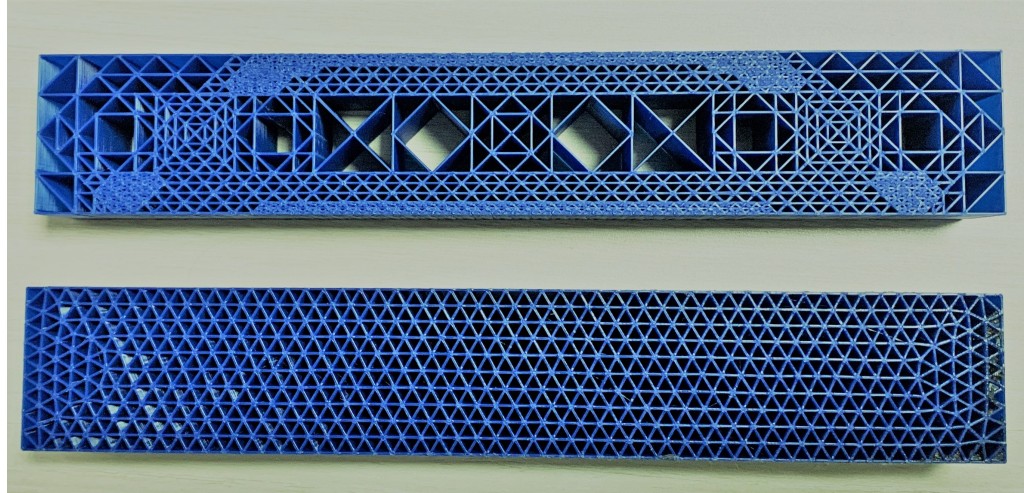

**Figure 15.** Printed parts with infill density of 41% having optimised infill design (**top**) and uniform infill design (**bottom**).

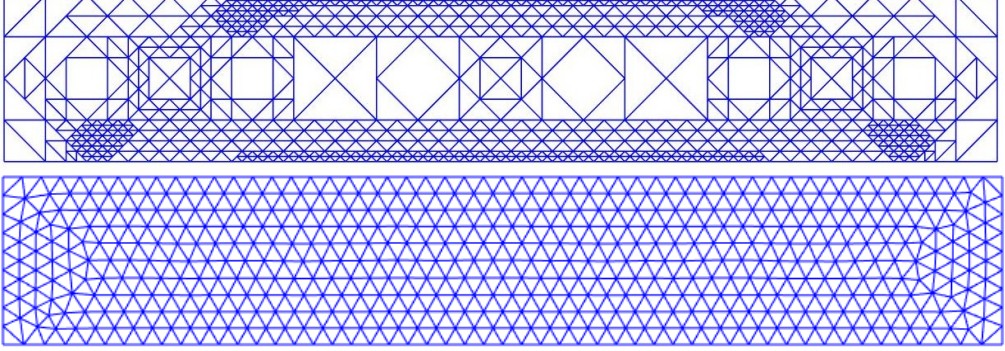

**Figure 16.** Part with optimised infill design (**top**) and uniform infill design (**bottom**).

*4.2. Comparison of Parts with Same Infill Density of 26%*

Similarly, in Figure 17 and Table 7, a comparison is illustrated between the part with uniform infill to the part with optimised infill with same infill density of 26%. Both in terms of maximum displacement (0.8 mm) and strain energy (156.28 mJ), the part with optimised infill using the refinement technique has shown significantly better performance in terms of maximum displacement and strain energy. Thanks to continuous printing, manufacturing time for the part with optimised infill design is 3.08 h compared to the part with uniform infill, which is 4.16 h (see Figure 18).

In contrast, maximum von Mises stress ($\sigma_{vm}$) in the case where the part with optimised infill is higher, when compared to part with uniform infill. This is due to the same reason mentioned in the previous comparison.

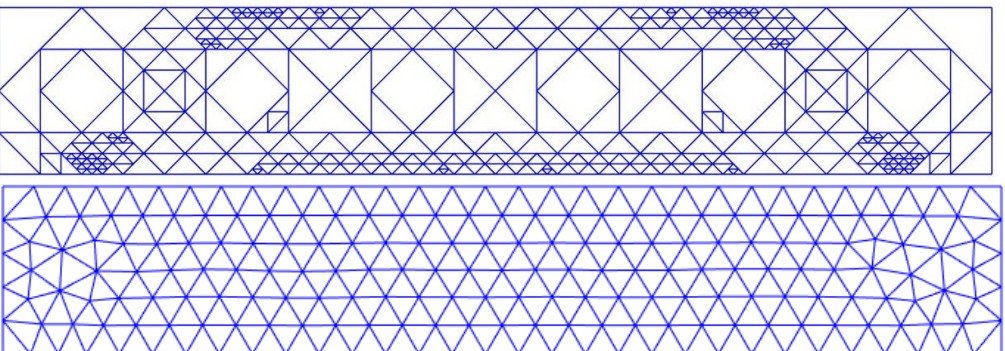

**Figure 17.** Part with optimised infill design (**top**) and uniform infill design (**bottom**).

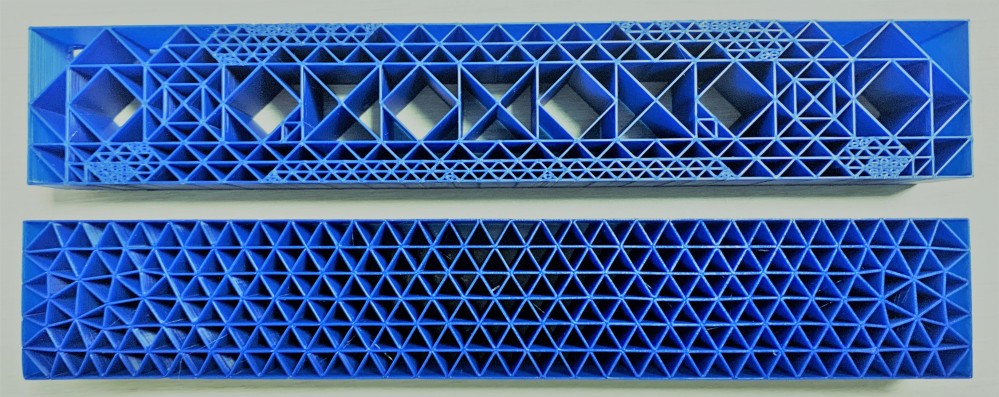

**Figure 18.** Printed parts with infill density of 26% having optimised infill design (**top**) and uniform infill design (**bottom**).

**Table 7.** Comparison of the part with uniform infill to the part with infill defined by the refinement process using threshold stress = 17.5 MPa and minimum length of a side of the infill design (L) $\geq$ 2.4 mm as stopping criteria.

|  | **Figure 17 (Top)** | **Figure 17 (Bottom)** |
| --- | :---: | :---: |
| **Infill Density (%)** | **26** | **26** |
| Von Mises Stress (MPa) | 36.4 | 25.1 |
| Maximum Displacement (mm) | 0.8 | 1.21 |
| Elastic Strain Energy (mJ) | 156.28 | 205.1 |
| Total mass of the part (g) | 32.1 | 32.8 |
| Time to print (hours) | 3.08 | 4.16 |

*4.3. Comparison of Parts with Same Infill Density of 22%*

Finally, Figure 19 and Table 8 portray a comparison of the part with uniform infill to the part with infill generated using refinement technique having same infill density (22%). The strain energy of a structure with uniform infill is then 225.9 mJ, whereas the part with infill generated using refinement technique using threshold stress as 25 MPa has a strain energy of 187.2 mJ, again indicating that the part with optimised infill is stiffer. Likewise, maximum displacement in the part with uniform infill is 1.33 mm, compared to 1.06 mm in the part having infill generated by refinement technique. Comparing manufacturing time, it

is evident that, by using the refinement technique (continuous printing), the manufacturing time is reduced by 0.8 h for the part with optimised infill compared to the part with uniform infill with almost same amount of mass (23 g) (See Figure 20).

Additionally, the maximum von Mises stress ($\sigma_{vm}$) in both cases is still different and is still higher in the part with optimised infill.

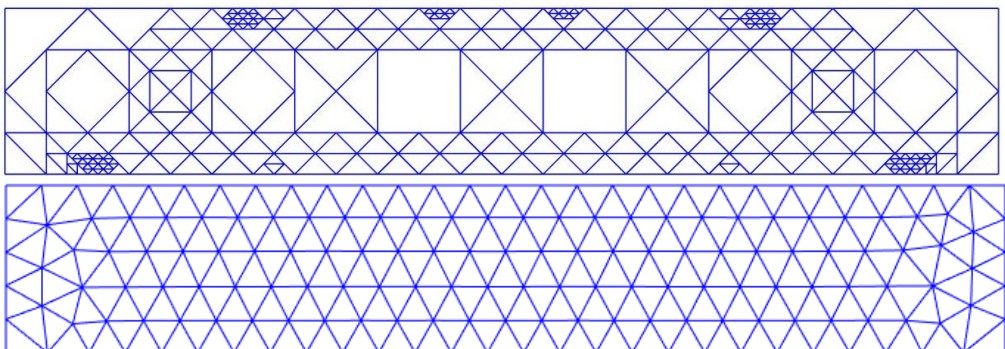

**Figure 19.** Part with optimised infill design (**top**) and uniform infill design (**bottom**).

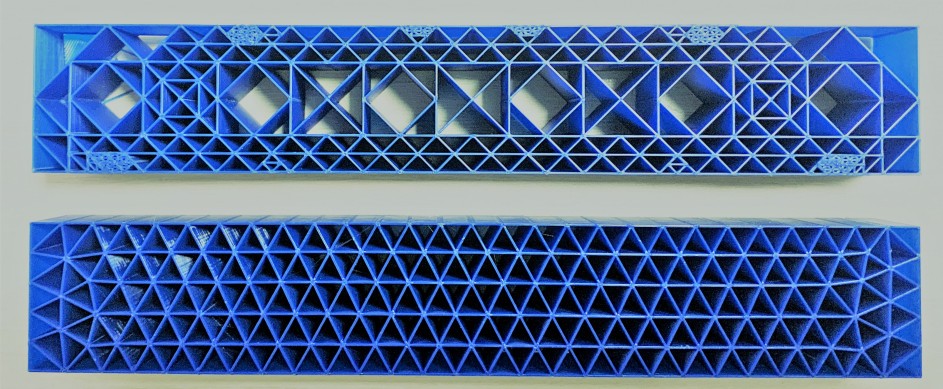

**Figure 20.** Printed parts with infill density of 22% having optimised infill design (**top**) and uniform infill design (**bottom**).

**Table 8.** Comparison of the part with uniform infill to the part with infill defined by refinement process using threshold stress = 25 MPa and minimum length of a side of the infill design (L) ≥ 2.4 mm as stopping criteria.

|  | **Figure 19 (Top)** | **Figure 19 (Bottom)** |
|:---:|:---:|:---:|
| **Infill density (%)** | **22** | **22** |
| Von Mises Stress (MPa) | 35.1 | 26.7 |
| Maximum Displacement (mm) | 1.06 | 1.33 |
| Elastic Strain Energy (mJ) | 187.29 | 225.9 |
| Total mass of the part (g) | 23.12 | 23.6 |
| Time to print (hours) | 2.83 | 3.63 |

These comparisons conclude and hold strong evidence that the refinement technique to generate infill brings benefits in terms of maximum displacement, strain energy and manufacturing time (continuous printing and zero airtime) over uniform infill distribution. The Printer used to manufacture the parts is Rise 3D V2 Plus (see Figure 21).

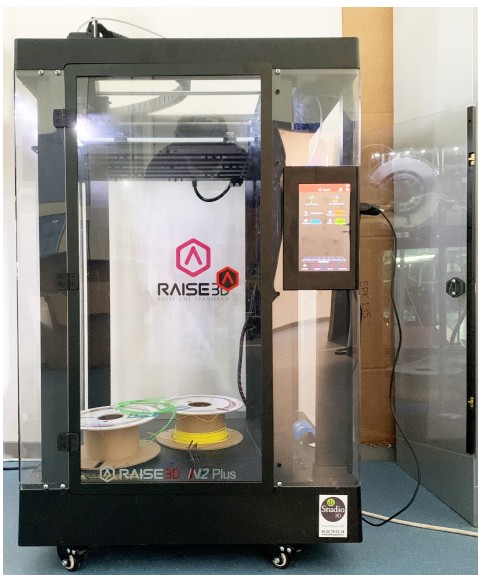

**Figure 21.** Rise 3D Printer V2 Plus used to manufacture the printed parts in Figures 15, 18 and 20.

## 5. Discussion

In the presented methodology, the refinement technique has been used for the generation of infill design. The ability to solve the problem of continuous printing has been achieved by using the Eulerian path. Using this refinement process, continuous printing is feasible while increasing the mechanical performance of the part at the same time. To illustrate the effectiveness of the proposed approach, an application using a case study has been presented and compared to the parts with uniform infill in Tables 6–8.

Focusing on the maximum absolute von Mises stress while comparing, it is noticeable that regardless of using the threshold stress ($\sigma_{thrld}$) as 10 MPa, 17.5 MPa and 25 MPa, the maximum absolute von Mises stress ($\sigma_{vm}$) of the final parts (35.3 MPa, 36.45 MPa and 35.1 MPa) are still not satisfying the stopping criteria of threshold stress ($\sigma_{thrld}$). Further continuing the iterative process is not possible due to the limitation of the length of the side of the infill, which is already below the given limit ($L \geq 2.4$ mm). According to this refinement technique, the objective was to subdivide infill design in such a way that continuous printing constraint is satisfied while increasing the strength of the part. In hindsight, like the work in [12,17], we suspect it could bring benefits to place the material as infill along the stress field direction during the refinement.

Some of the ideas that we will address concerning the present refinement methodology as future works are:

- Adding the capability to align the infill along the stress field by moving the nodes of the elements of infill design.
- Testing this refinement technique with more complex structures and loading conditions.
- Defining initial infill design of the part

### 5.1. Adding the Capability to Align the Infill along the Stress Field by Moving the Nodes of the Elements of Infill Design

Adding the capability to align the infill along the stress field by moving the nodes of the elements of infill design. The ability to align the beams in the stress direction could increase the strength of the part and decrease the maximum stress simultaneously. This could be possible with available optimisation techniques.

### 5.2. Testing This Refinement Technique with More Complex Structures and Loading Conditions

This process has been evaluated for two-dimensional loading cases. There is a need to understand more complex loading structures. This includes a 3D structure with complex

contours. The ability to find a standardised procedure for the complex part would enable us to compare along with a wide range of parts presenting different contours and infill designs.

### 5.3. Defining Initial Infill Design of the Part

The very first step of the refinement technique is to define the coarse initial infill structure of the part (see Figure 2). This initial infill structure for the case study part with 6.5% infill density (Figure 7) is arbitrarily defined. Hence, the final optimised part (see Figure 8) after the completion of the iterative process is dependent on the initial infill structure of the part chosen. This implies that the final infill structure of a part varies according to the initial infill structure of the part chosen. Hence, it raises the enigma to find the best initial infill structure that will result in the most optimised infill structure at the end of the refinement technique.

The opposite methodology could also be implemented by starting with full dense uniform infill as the initial infill of the part, then later removing the material as infill where it is not crucial during each iteration. This problem will be addressed in the next work.

### 6. Conclusions

This paper demonstrated a new way to define the infill design of 3D printed parts based on an FE Simulation and refinement technique. This process displayed the advantages of better material deposition as infill and creating a continuous path (leading to zero airtime while printing), while increasing stiffness at the same time and keeping the contour of the part constant. This process has been discussed in depth using a case study. Comparisons with the part containing uniform infill validated the effectiveness of the proposed methodology in terms of maximum displacement, strain energy and manufacturing time. While working on this project, a few more research areas have been brought to light. Some of them were discussed in the previous section and will be considered for our future work.

**Author Contributions:** All authors contributed equally. All authors have read and agreed to the published version of the manuscript.

**Funding:** This work in the paper has been part of OPTIFABADD project. This project is supported by the Department Council of Aube CD10 (France) and the European Regional Development Fund (ERDF).

**Institutional Review Board Statement:** Not applicable.

**Informed Consent Statement:** Not applicable.

**Data Availability Statement:** Not applicable.

**Conflicts of Interest:** The authors declare no conflict of interest.

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
