# Peer review of "Infill Design Reinforcement of 3D Printed Parts Using Refinement Technique Adapted to Continuous Extrusion"

_jmmp, doi:10.3390/jmmp5030071_

Round 1
Reviewer 1 Report
The manuscript is scientifically sound and reads well. The authors should address the following comments.
- Details of the finite element simulation analysis should be given or need to provide suitable references.
- Why were yielding and strain hardening not considered for structural analysis?
- Typos: von Mises (few typos throughout the text); page 14, line 333-334 is wrong; page 15: correct the sentences in lines 357, 367, and 394.
Author Response
- Details of the finite element simulation analysis should be given or need to provide suitable references.
For Abaqus finite element simulation, as a material, we chose PLA with Isotropic behaviour assumption. The simulation was implicit and Beam (B21: Timoshenko) element is used as a mesh element. The mesh size was 0.1 mm. Concentrated Load were applied of 250 N on each end.
- Why were yielding and strain hardening not considered for structural analysis?
Strain hardening is not considered because our focus was to work on the elastic limit (Elastic constitutive model). In future, we will be working on the plastic region and hence, on strain hardening. As we are working with PLA (Polylactic Acid) material, the yield strength of the PLA is 50 MPa. Therefore, we choose the Force on part as 250 N such that it never exceeds the yield strength.
- Typos: von Mises (few typos throughout the text); page 14, line 333-334 is wrong; page 15: correct the sentences in lines 357, 367, and 394.
These typos and error have been modified in the tex file of the article.
Reviewer 2 Report
The authors have proposed an iterative method to optimise the infill design for the 3D printing of components. The study looks interesting however before proceeding for publication, some comments need to be addressed.
- Conclude the literature in the introduction section from your point and include the research gap you addressed.
- Similar kinds of steps are followed for topology optimization. How is your method different from it?
- Include figures of Von-Mises stress distribution for each iteration obtained in Abaqus in Table 1.
- It will be good to include images of 3D printed components for optimised and uniform infill designs. Experimentally printing optimised and uniform infill designs, and including them in Figures 15, 16, and 17 will represent the feasibility of your study.
Author Response
Response to Reviewer 2 Comments
1. Conclude the literature in the introduction section from your point and include the research gap you addressed.
In the literature, many studies were conducted on topological optimisation. The purpose of topological optimisation is to find the most beneficial material deposition by changing the contour of the part. Whereas, infill optimisation addresses the same idea of the best material deposition while keeping the contour of the part constant.
For infill optimisation many studies has been carried using lattice structures. However, the importance of printing parameters was not taken into consideration while optimising infill. Therefore, our goal was to propose the infill optimisation technique taking into account the printing parameters such as minimum printable infill size possible and to reduce the manufacturing time of the final part using continuous extrusion.
**(Included in the modified version of the article)
2. Similar kinds of steps are followed for topology optimization. How is your method different from it?
Topology optimization is a process in which the design or contour of the whole structure is modified, and full material is used. In addition, previous research work on topological optimization neglected the printing parameters (E.g. Diameter of printing nozzle, G-code generation).
In infill optimization (refinement technique), the contour of the part remains constant, whereas the infill structure of the part is modified. While working on the technique printing parameters are also takes into consideration. Such as the minimum infill size of 1.2 mm is justified, according to the 3D printer nozzle diameter (0.4 mm) used. Finally, optimizing the printing time by implementing Eulerian path. Table 6, 7 & 8 validates the effect of our methodology (refinement technique) in terms of printing time, strength and displacement compared to part with uniform infill.
3. Include figures of Von-Mises stress distribution for each iteration obtained in Abaqus in Table 1.
The figures from Abaqus simulation for each iteration has been added in the article.
4. It will be good to include images of 3D printed components for optimised and uniform infill designs. Experimentally printing optimised and uniform infill designs, and including them in Figures 15, 16, and 17 will represent the feasibility of your study.
The picture of the printed parts has been attached to the modified article (Figure 16, 18 and 20). The figure of the printer is also added which was used to manufacture the part.

Round 2
Reviewer 2 Report
Authors have addressed all comments. No more comments from my side. The Paper can be accepted for publication.